:ᐧᐧᐧ: PLOS | ONE

# Does caching strategy vary with microclimate in endangered Mt. Graham red squirrels?

**Calebe Pereira Mendes**[ID][1]*, **John Koprowski**[2]

**1** Laboratório de Biologia da Conservação, Instituto de Biociências, Universidade Estadual Paulista "Júlio Mesquita Filho" (UNESP), Rio Claro, São Paulo, Brazil, **2** School of Natural Resources and the Environment, University of Arizona, Tucson, Arizona, United States of America

* calebepm3@hotmail.com

**Data Availability Statement:** All relevant data are within the manuscript and its Supporting Information files.

## Abstract

Food hoarding is a common behavior used by a variety of animals to cope with periods of low food availability. At the retreating edge of species' distribution, the stressful environment and unfavourable climate conditions may impose severe costs on hoarding behavior. Since relict populations are hotspots for evolution and adaptation, and considering that food hoarding behavior has a strong evolutionary basis, we decided to evaluate the occurrence of behavioral variability in the amount of food cached by the endangered Mount Graham red squirrel (*Tamiasciurus fremonti grahamensis*). We tested the variation in cache size in response to microclimate, soil relief, vegetation, food availability and squirrel sex. The number of pits excavated by squirrels to cache cones was used as a proxy of cache size and was affected by mountain slope aspect and density of trees. More pits were excavated in the northeast facing slopes. The density of trees negatively affects the cache volume on southwest slopes, but not on northeast slopes. The sex of the resident squirrel also affects the number of pits in the squirrel midden, with males excavating 47% more pits than females. Males and females also presented different responses to the mountain slope aspect, with females excavating more pits on northeastern slopes than on southwestern slopes, whereas male cache size did not vary with the slope aspect. Finally, the squirrel's caching behavior did not vary in response to midden microclimate variation, a result with possible implications for the survival of the Mt Graham red squirrels, given the predicted temperature increases in the region due to climate change.

## Introduction

In response to both predictable and unpredictable variation in food availability, several taxa (i. g. carnivores, rodents, corvids and birds of prey) make use of caching behavior to guarantee the food supply during periods of scarcity [1,2]. Caching behavior dampens the variations in food availability, "transferring" part of the resources from the period of abundance to be used during scarcity. Since the behavior is used by an array of different taxonomic groups, considerable variation exists in the type of food stored, amount, duration, substrate in which the food is cached, number and spatial distribution of caches [1,3]. For this reason, model organisms of

**Funding:** This work received a studentship from CAPEs (Coordenação de Aperfeiçoamento de Pessoal de Nível Superior - https://www.capes.gov.br), process number: BEX 0659/15-0 to CPM. The funder had no role in study design, data collection and analysis, decision to publish, or preparation of the manuscript.

**Competing interests:** The authors have declared that no competing interests exist.

several different taxa are needed to explore the general patterns and ecological implications of cache behavior.

Squirrels have long served as model organisms in the study of caching behavior in mammals, and have helped us to understand the cognitive mechanisms behind animal decision making, including which criteria are used to decide whether to consume or cache a food item [4] and how far to move a food item before caching it [5]. As result, the observed general pattern is that large food items are preferable options to cache [4] as well as items which are less susceptible to spoilage [3]. More valuable food items are usually dispersed farther from the food source to avoid conspecific pilferage [4], and feeding efficiency is an important factor for rodents, shaping not only their behavior but also morphology and evolution [6,7].

The North American red squirrel (*Tamiasciurus hudsonicus*) uses both larder and scatter-hoarding strategies according to cache site conditions [8]. The cache size is also adjusted to match future energy demand and differs between sexes, due to the differential energetic investment in the mating strategies [9]. For red squirrels, cache behavior is important for winter survival, but is also energetically expensive and time consuming [9], exposing the animal to predation risk [10]. The cache size must guarantee winter food security, but should not be large to the point where food is lost to spoilage [11]. Since the red squirrel cache depends on cold and moist condition for preservation [12], the behavior cost-benefit is also affected by the climate.

For populations at the edges of the species distribution, the harsh climatic conditions, often exacerbated by climate change, can severely disrupt the cache behavior effectiveness by increasing food spoilage [13]. Although disruptions in the cache behavior can be sometimes compensated by behavioral shifts, these behavioral shifts can also drive populations to extinction [13]. Indeed, behavioral shifts in response to changes in environmental factors, both due to natural or anthropogenic causes, can be benefic and allow the populations to persist and thrive in adverse environments, but can also be counteradaptative and create evolutionary traps, which can drive populations to extinction [14,15]. Since relict populations at the edges of the species distribution were gradually exposed to what is now a challenging and changing environment, and were selected from what once was the species core population, they are evolutionary hotspots [16,17], and therefore, excellent models to explore the effect of behavioral shifts on population survival.

This way, the relict population of red squirrels (*Tamiasciurus fremonti grahamensis* J. A. Allen, 1894) in the Mt. Graham, Arizona-U.S.A., presents an extraordinary opportunity to explore the occurrence of behavioral shifts in response to a harsh environment. The Mt. Graham red squirrel is a subspecies in the North American red squirrel complex (*Tamiasciurus hudsonicus*, *T. fremonti*) [18] isolated from the species core distribution for at least 11 million years [19]. The only population of this subspecies, with around 274 individuals during the study period [20], is severely threatened by the low habitat quality, insect outbreaks, exotic species and human activities [19]. Forest fires are also a threat, with the population being reduced to about 35 individuals in 2017 due to a large forest fire, and recovering to 67 squirrels in 2018 (J Koprowski, personal communication, 2018). In addition to all of these threats, this is the southernmost population of the species and is directly threatened by climate change, as they rely on cold and humid climate for the preservation of cones collected during the fall and consumed during the winter [12,21], and the region mean temperature is predicted to increase 1.5°C until 2100 [22].

Since behavior is among the most plastic traits of a species and modulates the species interactions with the environment, behavioral shifts are usualy the first adaptative reponses to appear in response to environmental challenges [23]. The main objective of this study was to evaluate the occurrence of behavioral shifts in the caching behavior of the Mount Graham red

squirrel (*Tamiasciurus fremonti grahamensis*) in response to microclimate variation. More specifically, we evaluated: 1- If climatic variables have effects on the squirrel's cache size, since climate change can interfere in the cost-benefit of food caching behavior [1] and a fine-tuning of the cache volume reduces energy waste [11]; 2- If relief variables such as altitude, slope declivity and aspect, have effects on the squirrel's cache size, since these relief variables affect the local microclimate [24]; 3- If food abundance has effects on the squirrel's cache size, since a higher food availability reduces the cost to find and transport the cones to the cache, increasing the overall cost-benefit of the cache behavior.

Based on these assumptions, we expect to observe bigger caches in colder areas, where the conditions to cone preservation is better and spoilage is expected to be reduced, increasing the cost-benefit of the behavior. We also expect to observe bigger caches in areas with colder microclimates, such as in higher elevations and in slopes facing the northeast. Finaly, we expect to observe bigger caches in areas with higher cone availability.

## Materials and methods

The study was undertaken in the highest elevations of Mount Graham, in the Pinalenō Mountains, Arizona, USA, between 2770 and 3270 m of altitude, an area with rugged relief, which promotes considerable microclimatic variation in a small geographic area, as part of the Madrean Archipelago. The Madrean Archipelago is a mid-latitude sky island complex, where temperate zone species are surrounded by a low altitude inhospitable matrix. The montane islands have high biodiversity, each one with its own set of relict populations that have persisted since the end of the last glaciation 10 to 15 k.y.a [21,25]. In fact, with the increase of ~5 °C in mean annual temperature since the Last Glacial Maximum, 19 to 23 k.y.a. [26] and the consequent poleward displacement of the distribution of the species, the Madrean Archipelago become the actual distribution edge of at least 80 species, including the red squirrels [25]. The study was conducted under permits from the Arizona Game and Fish Department (permit number SP696903), the US Fish and Wildlife Endangered Species (permit number TE041875), and the University of Arizona's Institutional Animal Care and Use Committee (permit number #14–504).

To evaluate the effect of climate on Mt. Graham red squirrel caching behavior, we selected 40 occupied middens distributed along different altitudes, relief, and mountain aspect. Since the subspecies' geographic range is extremely restricted, these middens were enough to cover the local habitat variability. We visited each midden 6 times between the fall and winter of 2015 to measure variables of weather, relief, plant structure and estimate the volume of cones stored by the resident squirrels (response variable). During the visits, we also recorded the sex of the resident squirrels, by visual inspection using binoculars.

To estimate the volume of cones cached by each squirrel, we took advantage of the fact that the animals do not cache the cones loose in the scale pile but deposit them into holes, here called "pits", excavated in the scale pile or in the soil [12]. Thus, we used the number of pits, counted during the cache season, as proxy of the volume of cones cached in each midden. This method follows Gurnell [27], but in the present study we did not excavate the pits to count the number of cones inside the pits, since it would disturb the midden of these endangered squirrels. The cache size estimates occurred between August 24 and September 12, an interval short enough to avoid any bias caused by the accumulation of cones along the cache season (LM, $\beta$ = 0.03758, t = 1.726, p-value = 0.096). We also recorded the presence of dead logs near middens, because they are occasionally used to store cones [12], but since the presence of logs had no detectable effects on the preliminary results, we excluded the variable from the analysis.

For the weather variables, we use a handheld weather meter (Kestrel 3000) to measure the air temperature (˚C) and relative humidity (%), and a digital soil thermometer (HANNA HI45-30) to record soil temperature, the temperature inside of the scale pile, and inside of the pits (average value obtained from 3 randomly selected pits for each midden). Since we could measure only one midden at a time, the dial temperature variation, as well as the variation along different days, could easily mask any difference in middens microclimate. To make the measurements comparable, we used the data recorded by a meteorological station located at the top of the mountain (MGIO-Mt Graham Summit KAZSAFFO4) and by a data-logger (HOBO U23 Pro v2) buried in the ground as a baseline to allow comparison between middens. The weather station was used as baseline for the air temperature and humidity whereas the buried data-logger was used as baseline for the temperatures of soil, pits and scale piles. Since both baselines are fixed in space, by subtracting the baseline values from the values recorded in the middens, taken at the same time, the resultant value is the difference between the midden and the baseline, which is a value comparable between middens. This way, the temperature and humidity variables are the mean of the values measured and corrected, for each midden, during the 6 visits.

Autumn is a critical period for the storage of cones because it is warmer and drier compared to the microclimate under the snow over winter, thus it is important to keep the cones in cool and humid conditions to avoid spoilage before the snow fall. Since the middens are usually cooler than the soil and air temperature during the fall (mostly due to evaporative heat loss) we calculated an additional variable called "pit cooling effect". This variable was calculated by subtracting the air temperature from the pit temperature, and represents how much cooler is the pit interior compared with the external environment.

Since some species are reported to cache more in areas with longer snow cover periods [2], and considering that memory and learning are important factors that modulate species behavioral [28], we decided to test the effect of snow cover duration on cache size. To do so, we capitalized on the fact that in 2015 all middens were covered by snow on December 12, and so we used the percentage of soil still covered by snow within a 5-m radius of each midden between 10 and 12 of March of 2016 as a proxy for the speed of snowmelt at each midden. We decided to use data from the 2015–2016 winter based on the premise that snow melting speed does not vary randomly along the mountain, but follows a pattern defined by relief and microclimatic factors [29]. Thus, a total of seven microclimatic variables were tested in the present work (Table 1 and S1 Dataset).

To record the topographic relief variation, we used a GPS unit to record the altitude of each midden, a clinometer to record the declivity and a compass to record the aspect. For vegetation variables, between between 10 and 12 August, we used a densitometer to record the forest cover for each midden and recorded the arboreal community with diamater at breast height (DBH) $\geq$ 10 cm by using 4 transects of 30x5 meters, starting from the midden in the four cardinal directions. The density of live conifers in the transects was used as a proxy for the availability of cones near each midden. This proxy is based on the premise that the mean cone production of the conifer species did not vary across the study area, which was tested and confirmed by the Moran's I test for spatial autocorrelation, using data from cone counting for 73 trees of the 3 species that produced cone crops in 2015 (Engelmann spruce: n = 33, p-value = 0.597; Douglas fir: n = 35, p-value = 0.59; Ponderosa pine: n = 5, p-value = 0.269), nor was correlated with mountain aspect, which was tested and confirmed by linear models (Engelmann spruce: $\beta$ = 0.694, t = 1.109, p = 0.276; Douglas fir: $\beta$ = 0.001, t = 0.389, p = 0.7; Ponderosa pine: $\beta$ = -0.008, t = -0.695, p = 0.537). From the tree transect data, we also calculated the tree density within the midden vicinity, density of live trees, density of dead trees,

**Table 1. Explanatory variables.**

| Variables | Type | Unit |
|---|---|---|
| Altitude | Relief | Meters |
| Declivity | Relief | Degrees |
| Aspect* | Relief | Bearing |
| Soil temperature* | Microclimate | ˚C |
| Air temperature* | Microclimate | ˚C |
| Air humidity | Microclimate | ˚C |
| Pit temperature | Microclimate | ˚C |
| Scale pile temperature | Microclimate | ˚C |
| Pit cooling effect | Microclimate | ˚C |
| Snow melt speed | Microclimate | Percentage |
| Forest cover | Forest structure | Percentage |
| Tree density* | Forest structure | Ind/ha |
| Live tree density | Forest structure | Ind/ha |
| Dead tree density | Forest structure | Ind/ha |
| Live conifer density | Food availability | Ind/ha |
| Engelmann spruce | Food availability | Ind/ha |
| Douglas Fir | Food availability | Ind/ha |
| Ponderosa pine | Food availability | Ind/ha |
| Mushroom productivity | Food availability | Grams |
| Resident squirrel sex | Sexual | Binary |

List of tested explanatory variables with the respective units. The principal explanatory variables, marked by a *, were selected by PCA analysis and were used to create models for subsequent model selection analysis.

density of Englemann spruce, Douglas fir and Ponderosa pine, which are the main tree species that produced cones.

Between 3 and 7 August, we also estimate the productivity of mushrooms, an alternative food resource used and cached by squirrels [19,30], and their availability may interfere with the behavior of the animals. Mushroom productivity, recorded in grams, was estimated through 4 transects of 10x1 m, starting from the midden in the four cardinal directions. All mushrooms of all species known to be consumed by the squirrels [30,31] were collected and weighed.

To analyze the data, we verified the normality of the variables, using the Shapiro-Wilk test, and normalized the variables when needed. The response variable, number of pits, was log transformed to fulfill normality requirements, and the slope aspect, a circular variable, was tested for uniformity using a Kuiper uniformity test [32]. To allow the slope aspect variable to be analyzed together with the other linear variables, we linearized the variable by dividing it along the northeast-southwest axis (bearings 45˚ and 225˚, where north is 0˚\360˚), and transfering the records in the northwest semicircle to the southwest semicircle. This way, the resultant semicircle can be treated as a linear variable, ranging from bearing 45˚ to 225˚, where 45˚ represents a place where the soil slope is facing the northeast, whereas 225˚ represents a place where the soil slope is facing towards southwest. We decided to use the northeast-southwest axis based on visual and preliminary inspections of the raw data, which pointed to maximal variation along the northeast-southwest axis.

Due to the large number of explanatory variables, to avoid type I statistical errors, we divided the variables into biotic and abiotic and used Principal Component Analysis (PCA) to identify and delete the variables with high similarity within each group [33]. In this process,

we used the PCA results for each group, selecting the more dissimilar variables based on the dissimilarity values within the first two dimensions, testing whether the variables were correlated with other similar variables, and then, deleting the correlated variable with smaller dissimilarity values. We performed a PCA again with the remaining biotic and abiotic variables, repeating the process to keep only the most dissimilar explanatory variables without correlations. After reducing the number of explanatory variables, we used a multiple competing hypothesis approach [34] to evaluate the effect of these remaining variables on the number of pits excavated by the squirrels.

For the competing hypothesis approach, we created a set of Generalized Linear Models (GLM), in which each model uses a different combination of one or two explanatory variables to explain the response variable (number of pits). For the model with two explanatory variables, we tested for both summatory and interaction effects between the explanatory variables, however, since interaction models performed poorly when compared to summation models, they were removed of the analisys. We did not test models with more than two variables due to the modest sample size of the dataset. A null model was created by using aleatory generated numbers as an explanatory variable, and the Akaike Information Criterion corrected to small sample size was used to compare the models [34]. For each model, we calculated the wAICc, a parameter of the relative likelihood, and the ΔAICc, a parameter of relative difference between models. Models with ΔAICc < 2 were considered equally plausible. To reinforce the analysis, we calculated frequency πi, which is a bootstrap method to calculate the frequency in which a model *i* is selected as the best model in the set in 10000 random resamples of the dataset [34].

Finally, due to differences in reproductive strategy between males and females, the peak of reproductive energy expenditure occurs in early spring for males and late spring for females [9]. Considering that in early spring, the males rely on cached food to maintain energy demanding reproductive activities, we tested the effect of sex of the resident squirrel on the cache size using Student's t-test and linear model regression. All statistics were performed using the R software [35] software, using the packages "circular" [36], "bbmle" [37], "FactoMineR" [38], "akima" [39], "plotrix" [40] and "ape" [41].

## Results

From the 40 selected middens, 12 were excluded due to migration/death of the resident squirrel during the study. From the 28 remaining, 15 were occupied by males, 11 by females and 2 animals of unknown sex. The 28 remaining middens were evenly distributed in all aspects of the mountain slopes (Kuiper's Test of Uniformity, p > 0.15).

The PCA analysis reduced the 19 initial explanatory variables (excluding the resident squirrel sex, which is a binary variable) to only 4 principal explanatory variables, which are: air temperature, soil temperature, aspect and the density of trees in the midden vicinity (S1 Fig). The other 3 vegetation variables, 2 relief variables, 5 climate variables and 5 food availability variables were removed from the analysis due to their correlation with at least one of the four more dissimilar variables. Based on the four selected explanatory variables, eleven models were created, including the null model, and compared using the Akaike Information Criterion method (Table 2). The model with highest explanatory power included the summed effects of relief aspect and density of trees as predictor variables (Fig 1), with the middens located in the northeastern slopes containing more pits than the middens in the southwestern slopes. At the same time, in the southeastern slopes, the middens surrounded by more trees contained fewer pits than the middens with less trees in its vicinity. This negative correlation between density of trees and number of pits was not observed in the northeast-facing slopes.

**Table 2. Variables affecting red squirrel cache size.**

| Models | ΔAICc | Df | wAICc | πi |
|---|---|---|---|---|
| GLM10: NP ~ Aspect + Density of Trees | 0.0 | 4 | 0.7161 | 0.599 |
| GLM3: NP ~ Aspect | 3.4 | 3 | 0.1337 | 0.061 |
| GLM8: NP ~ Aspect+ Air Temperature | 4.2 | 4 | 0.0865 | 0.220 |
| GLM6: NP ~ Aspect+ Soil Temperature | 5.3 | 4 | 0.0518 | 0.035 |
| GLM5: NP ~ Soil Temperature + Air Temperature | 10.3 | 4 | 0.0041 | 0.058 |
| GLM4: NP ~ Density of Trees | 11.5 | 3 | 0.0022 | 0.001 |
| GLM9: NP ~ Air Temperature + Density of Trees | 11.6 | 4 | 0.0021 | 0.017 |
| GLM7: NP ~ Soil Temperature + Density of Trees | 12.4 | 4 | 0.0015 | 0.003 |
| GLM1: NP ~ Soil Temperature | 12.9 | 3 | 0.0011 | 0.001 |
| GLM2: NP ~ Air Temperature | 14.5 | 3 | <0.001 | 0.000 |
| GLM0: NP ~ Null (aleatory generated values) | 15.1 | 3 | <0.001 | 0.000 |

Results of the model selection for Number of Pits (NP) excavated by Mt. Graham red squirrels. We fitted generalized linear models and calculated the corrected Akaike relative difference (ΔAICc), relative likelihood (wAICc) and a bootstrap model selection frequency (πi) to all models. Models with ΔAICc ≤ 2 were considered as equally plausible. The πi parameter was calculated based on 10.000 permutations.

The best model explained 44% of the observed variation in midden number of pits and received a wAICc = 0.7162, a frequency πi = 0.59, by far the most plausible model (Table 2). Aspect was clearly the most important variable, since it was also present in the four models with higher explanatory power, while the tree density variable was, by itself, a weak variable and obtained only the sixth position in the rank of better models.

Due to its categorical nature, the resident squirrel sex was not included in the PCA and competing hypothesis analysis, instead, we divided the middens according to the sex of the resident squirrel and compared the two groups with student t-tests. We found differences in the number of pits excavated by males and females ($t$ = 2.59, df = 23.78, p = 0.016), with males digging 47% more pits than females (Fig 2). We found marginal differences in the aspect of middens of males and females ($t$ = -0.105, df = 20.196, p-value = 0.054), with more male squirrels being found on slopes with a northeastern aspect and more females in the southwestern facing areas (Fig 3). We found no evidence of an aspect effect on the cache behavior of males ($\beta$ = -0.0006, t = -0.603, p = 0.556), whereas for females, the effect of the aspect was strong ($\beta$ = -0.359, t = -7.877, p $\ll$ 0.01), with females on northeastern slopes digging more pits than the females on the southwest facing areas (Fig 4).

## Discussion

We found evidence to support our hypothesis about the existence of behavioral variation in the cache behavior of Mount Graham red squirrels, so that the number of pits excavated by these rodents changes according to the site aspect, density of trees and the sex of the resident squirrel. The data did not support the existence of temperature related behavioral variation, as squirrels displayed no response to any of the tested temperature variables. We found support for the existence of behavioral shifts in response to topographic relief, with middens in the northeastern slopes of the mountain containing more pits than in the southwestern slopes. This response was driven by females that strongly responded to aspect, whereas males did not. We found no support for behavioral shifts in response to food availability.

The absence of a response in the squirrel's cache behavior to altitude, with the variable even excluded during the PCA, is not surprising considering that altitude per se (i.e. vertical

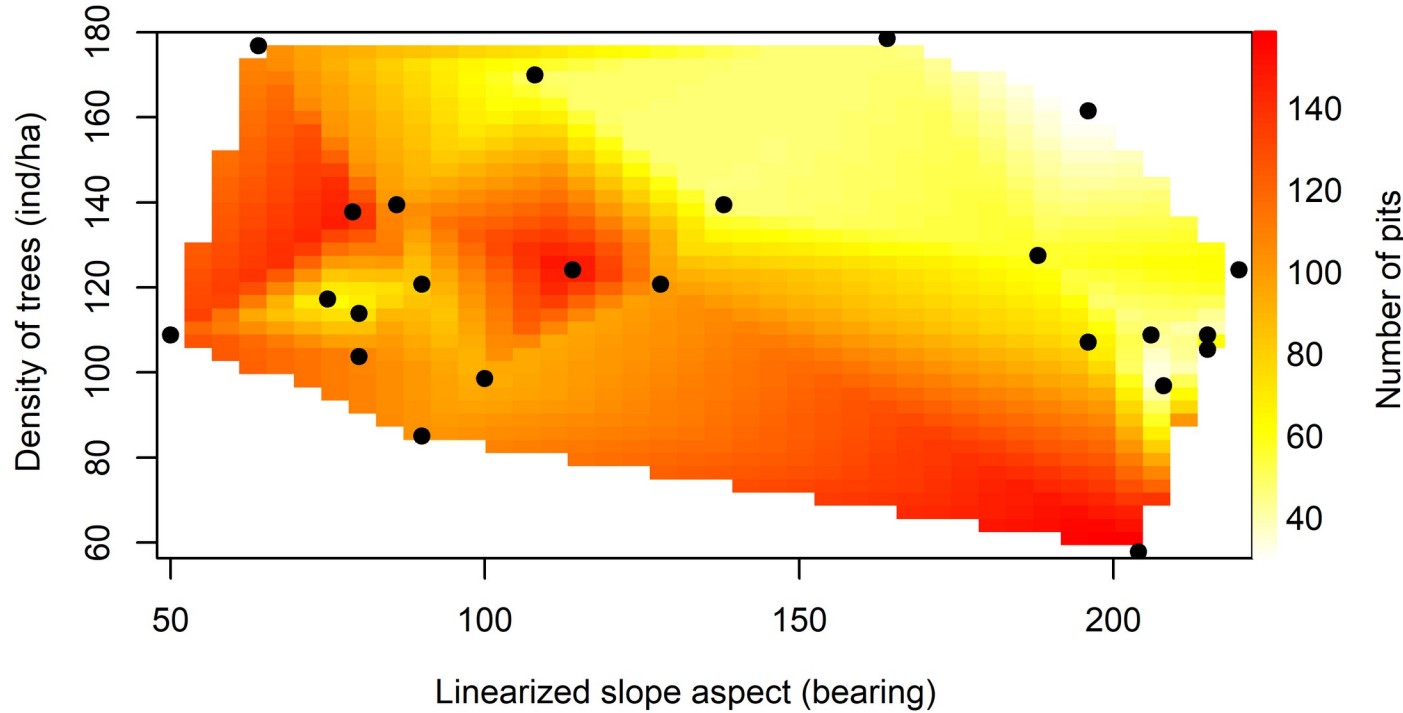

**Fig 1. Cache size in response to site aspect and tree density.** Predicted number of pits excavated by Mt. Graham red squirrels in response to the site aspect (linearized in the x-axis), and density of trees (y-axis), based on the best fitted model. The black points represent the sampled middens.

distance between a point and sea level) did not have causative effects on biological systems. Instead, altitude must be considered as a strong proxy of other environmental variables, such as barometric pressure, temperature and humidity, which in turn have effects on biological systems [24]. Considering that temperature and humidity were also poor predictors of squirrel cache behavior, it is not a surprise that altitude was excluded by the PCA.

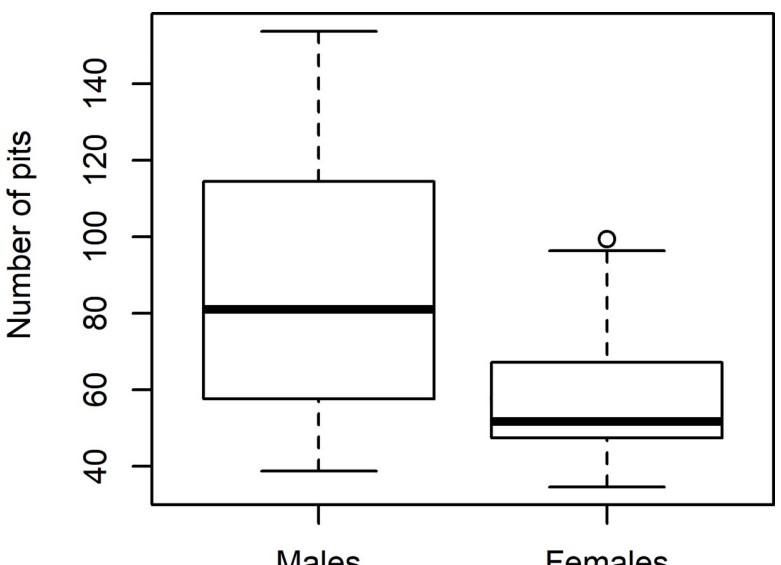

**Fig 2. Cache size of males and females.** Differences in the number of pits excavated by male and female red squirrels on Mt. Graham, Arizona, USA.

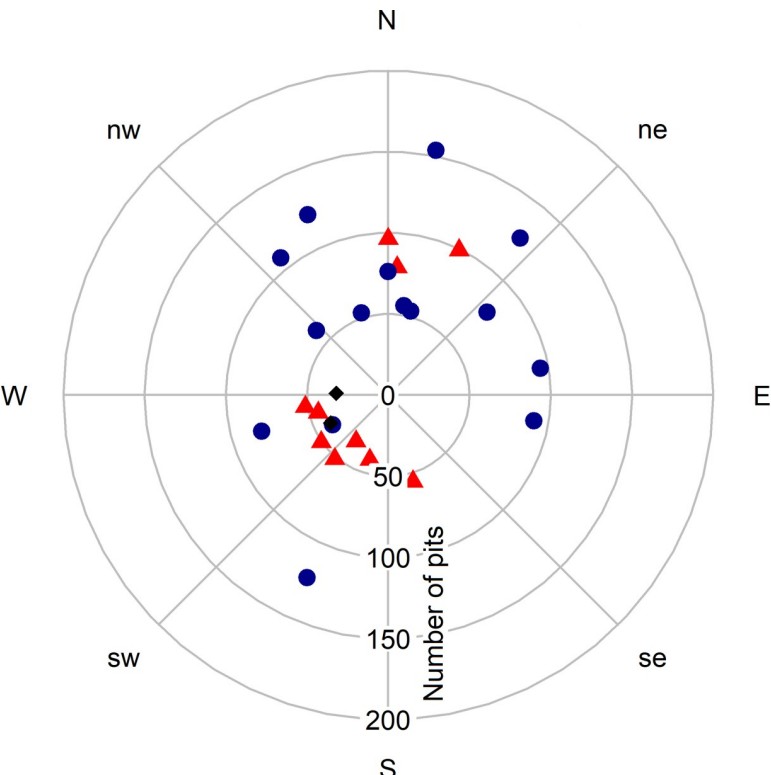

**Fig 3. Cache size along the mountain aspect.** Circular distribution of the sampled middens along the mountain aspect. The radial axis represents the number of pits excavated by each resident squirrel. The distance between points in the picture does not represent geographical distances at the study site. The sex of the resident squirrels is indicated by blue points for males, red triangles for females and black diamonds for squirrels with unknown sex.

The absence of a response evidenced in the squirrel's cache behavior to variation in the temperature along the mountain is surprising given how important temperature is to cone preservation [12,42] and thus to squirrel survival. However, several factors could interfere and/or

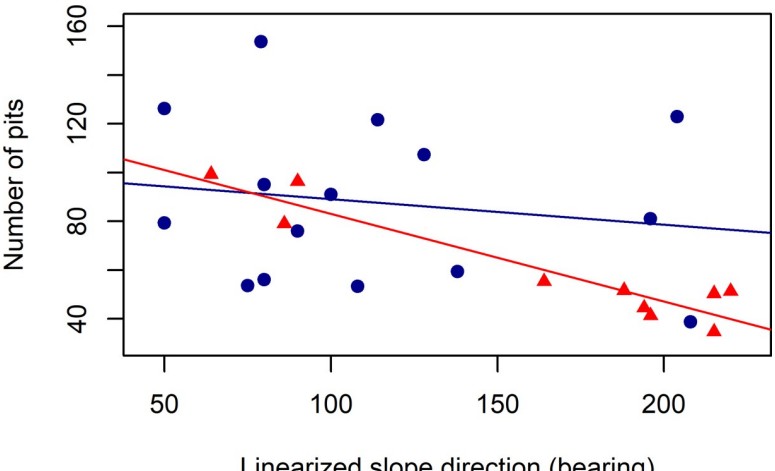

**Fig 4. Differences in the response of males and females to site aspect.** Regression of the number of pits excavated by males (blue points) and females (red triangles) in response of the aspect (linearized in the x-axis). For the description of the linearization process of the aspect variable, see the methods section.

preclude the emergence and maintenance of such behavioral responses. First, mean air temperature varied only 4.7˚C between middens, it is possible that squirrels simply lack the capability to detect such small variation and thus, do not respond to these small temperature variations, independent of its importance to cone preservation. Even if squirrels are not capable of perceiving this variation in mean temperature, increases of just a degree or two are enough to impact cone preservation (J Koprowski, unpublished data), and a minimum increase of 1.5˚C is expected due to climate change before 2100 [22].

Another possible explanation is that the squirrels did not respond to the mean temperature, but to the occurrence and duration of extreme temperature events, such as observed in the Canada jay (*Perisoreus canadensis*) [13]. For food preservation, one or few events of extreme temperatures may cause cache spoilage through frost thaw or, in the case of red squirrels, by cone dehydration [12,13]. However, extreme temperature events were not detectable by the current experiment design, and therefore, a more specific experiment would be required to clarify this issue. This is a question with high conservation value, since it relates to whether the Mt. Graham red squirrels have behavioral ways to deal with the climate change predicted to the region [22].

The present study observed the effect of the site aspect, density of trees and sex of the resident squirrel on cache size, however, it was not possible to completely isolate and precisely estimate the effect of each one of these three variables on the animals' cache behavior. The isolation and evaluation of these three effects was hindered by the unexpected unequal distribution of males and females in the montain slopes, along the Notheast-Southwest axis. The loss of 30% of the middens due to squirrel mortality/migration during the sampling period further reduced the sample size and therefore limiting the statistical options available for data corrections. To precisely estimate the effect of aspect, density of trees and squirrel sex on cache size, a study with standardized sample size for males and females along the mountain aspect is recommended, using an initial sample size big enough to compensate the low annual survival rate of Mt Graham red squirrels [20].

Althouth the exact mechanisms underlying the observed results are still not clear, there are some possible explanations that fit the observed data. A possible mechanism driving the effect of both aspect and density of trees on cache size is the incidence of solar radiation. In the northern-hemisphere, the southern mountain slopes receive sunlight at a more perpendicular angle than the northern slopes, and the forest canopy blocks more solar radiation from reaching the ground when the light came from a less perpendicular angle, such on the northern slopes [43]. Together, aspect and density of trees cause the northern slopes to be more shaded than southern slopes. This way, solar radiation might explain why cache size varies along the northeast-southwestern axis of the aspect, as observed for female squirrels, but its consequences on the squirrel cache are not clear.

Another possible mechanism underlying the observed differences in responses of males and females to tree density is related to the individual's perception of predation risk. Differences in boldness, vigilance and risk perception between sexes are a relatively common phenonema [44], and vigilance activities do interfere with foraging efficiency [45]. Since avian predation is a significant cause of mortality in the Mt. Graham squirrel population [10,20] and females have a higher annual survival rate than males [20], it is possible that tree density has different effects on risk perception for males and females, affecting their respective cache efficiency, effort and therefore cache size.

Our results document the existence of cache related behavioral variation in response to the topographic relief, density of trees and individual sex, in an endangered red squirrel subspecies. Although the mechanism by which these variables affect squirrel cache behavior are still not clear, it may be unrelated to site temperature, as would be usually expected based on the

literature. Since temperature is important to cone preservation [12], and the Mt. Graham red squirrels did not appear to behaviorally respond to temperature, the future increase in temperature may turn the Pinalenõ Mountains into unsuitable forest for these red squirrels. In the face of already unavoidable climate change, sky island complexes, such as the Madrean Archipelago [25,46], can be used as natural laboratories, providing us the opportunity to better understand how species characteristics, including behavior, affects its persistence capability. This may also allow us to develop and improve our conservation methods to protect populations on the edge of their distributions and on the edge of extinction.

## Supporting information

**S1 Dataset. Complete dataset.** .CSV file containing the data used in the present study.
(CSV)

**S1 Fig. Principal component analysis.** PCA, showing the multivariate variation of the biotic (a) and abiotic (b) explanatory variables. Variables with similar variation were tested for correlation by the Pearson's correlation coefficient, when correlations were found, the most dissimilar variable was kept while the other was discarded. The resulting biotic and abiotic variables were put all together and the process was repeated (c). In the end of the process, only the four variables (tree density, soil temperature, air temperature and aspect in a Northeast-southwestern axis) remained and therefore were used in the model selection.
(TIF)

## Acknowledgments

We would like to thank to the Conservation Research Laboratory of the University of Arizona, Arizona Agriculture Experiment Station and the USDA Forest Service for logistic and financial support, to the Coordenação de Aperfeiçoamento de Pessoal de Nível Superior (CAPES) and Universidade Estadual Paulista "Júlio de Mesquita Filho" (UNESP-RC) for providing a one-year studentship at the study site, and to the Conselho Nacional de Desenvolvimento Científico e Tecnológico (CNPq) for a studentship in Brazil.

## Author Contributions

**Conceptualization:** Calebe Pereira Mendes, John Koprowski.

**Data curation:** Calebe Pereira Mendes, John Koprowski.

**Formal analysis:** Calebe Pereira Mendes.

**Funding acquisition:** Calebe Pereira Mendes, John Koprowski.

**Investigation:** Calebe Pereira Mendes, John Koprowski.

**Methodology:** Calebe Pereira Mendes, John Koprowski.

**Project administration:** Calebe Pereira Mendes, John Koprowski.

**Resources:** Calebe Pereira Mendes, John Koprowski.

**Software:** Calebe Pereira Mendes.

**Supervision:** John Koprowski.

**Validation:** John Koprowski.

**Visualization:** Calebe Pereira Mendes.

**Writing – original draft:** Calebe Pereira Mendes.

**Writing – review & editing:** Calebe Pereira Mendes, John Koprowski.

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
