## [Decision Letter · Decision Letter 0]

21 Aug 2019

PONE-D-19-16349

Does caching strategy vary with microclimate in endangered Mt. Graham red squirrels?

PLOS ONE

Dear Dr. Pereira Mendes,

Thank you for submitting your manuscript to PLOS ONE. After careful consideration, we feel that it has merit but does not fully meet PLOS ONE’s publication criteria as it currently stands. Therefore, we invite you to submit a revised version of the manuscript that addresses the points raised during the review process.

We have received comments from two reviewers and both of these reviewers have recommended minor revisions to the manuscript before it can be considered for publication. I agree with both of the reviewers' comments and in particular those of reviewer 2 that suggest that further justification and discussion is needed for a multifactorial analysis with your small sample size. I  recommend that you address this and the other comments carefully in your revision to the manuscript.

We would appreciate receiving your revised manuscript by Oct 05 2019 11:59PM. To enhance the reproducibility of your results, we recommend that if applicable you deposit your laboratory protocols in protocols.io, where a protocol can be assigned its own identifier (DOI) such that it can be cited independently in the future. For instructions see: http://journals.plos.org/plosone/s/submission-guidelines#loc-laboratory-protocols

We look forward to receiving your revised manuscript.

Kind regards,

Jesus E. Maldonado, Ph.D

Academic Editor

PLOS ONE

Journal Requirements:

Reviewers' comments:

Reviewer's Responses to Questions

**Comments to the Author**

1. Is the manuscript technically sound, and do the data support the conclusions?

Reviewer #1: Yes

Reviewer #2: Yes

2. Has the statistical analysis been performed appropriately and rigorously? 

Reviewer #1: Yes

Reviewer #2: Yes

3. Have the authors made all data underlying the findings in their manuscript fully available?

Reviewer #1: Yes

Reviewer #2: Yes

4. Is the manuscript presented in an intelligible fashion and written in standard English?

Reviewer #1: Yes

Reviewer #2: Yes

5. Review Comments to the Author

Reviewer #1: This manuscript explores a highly fascinating behaviour 'caching variation in response to environmental and individual factors (gender)' in an endangered tree squirrel that inhabits a mountain range at the edge of the species distributional range. The latter is critical as these populations are subject to different evolutionary drivers than populations at the core of the distribution and therefore of particular interest when it comes to understanding behaviours and adaptions in a changing world.

The manuscript is exceedingly well written and a very interesting read. My only comment is that given the aim of the authors, this manuscript establishes a baseline and as a concluding thought I would encourage the research group to repeat this study in the future once climate change effects have progressed to see how this is affecting caching decisions and conclusions drawn to date. This would also show if squirrels respond to extreme temperature events. In this context a control study area at the core of the distribution not subject to extreme temperature fluctuations would be helpful for comparison.

Specific comments

1. Line 55: the statement about distance here lacks context. Is it really distance per se or more an increase in predation risk which makes retrieving this item for other individuals not familiar with the precise location of the valuable item more risky?

2. Line 57: From a European perspective it would perhaps be helpful here at first mention of this species (as opposed to Mt Graham red squirrel above) to at least once in addition to the scientific name to say the North American red or pine squirrel (Tamiasciurus hudsonicus) to avoid confusion for general readers.

3. Line 64: should be 'depend'

4. Lines 68 and 71 - you have given Waite and Strickland 2006 here as full text when all other references are given as numbers - in your list it is at 42 and this should be changed.

5. Line 111 - I wonder if these "absolute" predictions need to be modified or set into context? I assume it is a given that predictions 1 and 2 (lines 107-110) are also subject to local cone (mushroom) availability per se (your variable of density of trees in midden vicinity?

6. Line 196 'varied' should be 'vary'

7. Lines 196 to 199 - as aspect is a critical factor in your later findings, did you include aspect as a variable here - did/would the sample of 73 trees allow this? One underlying assumption in your study is that all trees within midden vicinity are equal in terms of food availability and their density serves as a proxy. One might in principle expect aspect to matter in terms of cone production and this should be checked.

Reviewer #2: This paper describes an important evolutionary adaptation: behavioural variability in the amount of food cached by the endangered Mount Graham red squirrel (Tamiasciurus fremonti grahamensis). Authors tested the variation in cache size in response to microclimate variation, based on the following parameters: relief, vegetation (tree stand density), food availability and squirrel sex. Based on assumptions and predictions obtained from previous studies scientists expected that squirrels should locate bigger caches in areas:

1. Where the conditions to cone preservation are better and spoilage is expected to be reduced (colder areas).

2. With colder microclimates: in higher elevations and in slopes facing the northeast.

3. With higher cone availability.

To verify these hypothesis authors selected 40 middens occupied by squirrels distributed along different altitudes, relief, and mountain aspect. They visited each selected midden six times between the fall and winter of 2015.

Results of the study show, that much more pits were excavated in the northeast facing slopes. Surprisingly, the density of trees negatively affects the cache volume on southwest slopes only. The sex of the squirrel also affects the number of pits in the squirrel midden, males excavating more pits than females. Additionally, both sexes presented different responses to the mountain slope aspect (females excavating more pits on northeastern slopes, whereas male cache size did not vary with the slope aspect). Contrary to authors’ predictions the squirrel’s caching behaviour did not vary in response to midden microclimate variation.

However, although both the idea and conducted study are very interesting and worth publication, there are some aspects which require further explanation.

First of all, in my opinion (and as the authors mentioned in the discussion), the sample size may be not big enough for such multifactorial analysis. The second question is a type of distribution of studied middens – does it cover all habitat variability?

For me, the relationship between altitude, microclimate and the number of pits in the squirrel’s midden is the most important and missing information which may help us understand behavioural adaptation to the climate change. Authors should expect differences in caching squirrels’ behaviour in regard to differences in elevation gradient. In my opinion this aspect should be deeply analysed and discussed. Maybe squirrels territories were located at the same elevation – it should be disclosed. I am also wondering why authors analyzed differences between squirrels’ sexes? Did they expect any sex-related differences in catching behaviour and if so, there should be an explanation.

It would be good to introduce potential readers for studied species biology and describe briefly the subject of the study, including sexual differences, body mass and social system.

When you try to consider what causes could lead to the obtained results – the lack of behavioural reaction on micro-climate variability – the good way is to analyze the used methodology. It is possible that daily air temperature, which was measured in the meteorological station differs slightly from the daily air temperature in other places, like middens. The same could be applied to soil temperature monitored by data-logger between one (the same) place and other places (e.g. middens).

Other explanation which the author discussed in the manuscript is that the most important for catching behaviour was the maximum, not the average air temperature.

Abstract

P2 L22 - …” in response to climate”… - should be “in response microclimate”

Materials and Methods

P6 L131 – 134

“We visited each midden 6 times between the fall and winter of 2015 to measure

variables of weather, relief, plant structure and estimate the volume of cones stored by

the resident squirrels (response variable). During the visits, we also record the sex of the

resident squirrels.”

Could you please explain how did you record the sex of the resident squirrels?

It’s unclear – Did you measure all of these variables during each of the six visits? If not please add the information when did you measure each variable. As I understood the temperature in all middens was measured on the same day.

How did you choose three pits from one midden for temperature monitoring?

How accurate was your equipment for the weather variables (handheld weather meter Kestrel 3000) and a digital soil thermometer (HANNA HI45-30), it’s possible that low accuracy can mask differences in temperature and humidity.

Discussion

In this part of the manuscript, there is no explanation of why sex should generate differences in squirrel’s catching behaviour.

Additionally, the authors’ hypothesis about different levels in predation risk between places with high and low tree steam density as an explanation in differences in squirrels’ catching behaviour seems to be unconvincing. First of all, it is hard to believe that predation risk can vary between different mountain slope aspects.

6. PLOS authors have the option to publish the peer review history of their article (what does this mean?). If published, this will include your full peer review and any attached files.

Reviewer #1: Yes: Peter Lurz

Reviewer #2: Yes: Zbigniew Borowski

---

## [Author Response · Author response to Decision Letter 0]

2 Sep 2019

We thank the reviewers, for provide such a helpful feedback. Here are the responses to the questions.

Reviewer #1: This manuscript explores a highly fascinating behaviour 'caching variation in response to environmental and individual factors (gender)' in an endangered tree squirrel that inhabits a mountain range at the edge of the species distributional range. The latter is critical as these populations are subject to different evolutionary drivers than populations at the core of the distribution and therefore of particular interest when it comes to understanding behaviours and adaptions in a changing world.

The manuscript is exceedingly well written and a very interesting read. My only comment is that given the aim of the authors, this manuscript establishes a baseline and as a concluding thought I would encourage the research group to repeat this study in the future once climate change effects have progressed to see how this is affecting caching decisions and conclusions drawn to date. This would also show if squirrels respond to extreme temperature events. In this context a control study area at the core of the distribution not subject to extreme temperature fluctuations would be helpful for comparison.

Specific comments

1. Line 55: the statement about distance here lacks context. Is it really distance per se or more an increase in predation risk which makes retrieving this item for other individuals not familiar with the precise location of the valuable item more risky?

R: Indeed, the statement was unclear. We added some words to make the background clearer. But we didn’t add a complete explanation since it is provided in the cited manuscript.

Since squirrels usually cache food in the surroundings of the food source, farther places are safer in terms of conspecific pilferage. However, it takes more time and energy to travel farther before cache an item, increasing the general cost of the cache behavior by increasing the energy expended/energy retrieved (when the food is recovered) and reducing the energy cached/foraging time. This way, only more valuable food items worth the longer travel to a safer caching spot.

2. Line 57: From a European perspective it would perhaps be helpful here at first mention of this species (as opposed to Mt Graham red squirrel above) to at least once in addition to the scientific name to say the North American red or pine squirrel (Tamiasciurus hudsonicus) to avoid confusion for general readers.

R: Thank you for highlighting this issue. The words were added.

3. Line 64: should be 'depend'

R: We corrected the phrase grammar by changing the previous word (“caches” to “cache”).

4. Lines 68 and 71 - you have given Waite and Strickland 2006 here as full text when all other references are given as numbers - in your list it is at 42 and this should be changed.

R: The citations were corrected, making sure to meet the journal’s guidelines.

5. Line 111 - I wonder if these "absolute" predictions need to be modified or set into context? I assume it is a given that predictions 1 and 2 (lines 107-110) are also subject to local cone (mushroom) availability per se (your variable of density of trees in midden vicinity?

R: To avoid make unnecessarily complex predictions, we carefully created only three simple and direct predictions, each one with single predictor subject (1. Large scale temperature, 2. microclimate, 3. food availability). Yes, we indeed expected these predictions to interfere with each other, and this is the reason why we make sure to use non-excluding predictions (the three can be truth and have summed effects). We even tested these interactions in the methods to some degree, always being careful to choose a less demanding statistic approach due to the limited sample size.

6. Line 196 'varied' should be 'vary'

R: The word was changed.

7. Lines 196 to 199 - as aspect is a critical factor in your later findings, did you include aspect as a variable here - did/would the sample of 73 trees allow this? One underlying assumption in your study is that all trees within midden vicinity are equal in terms of food availability and their density serves as a proxy. One might in principle expect aspect to matter in terms of cone production and this should be checked.

R: Thank you for highlight this issue. We added the information about the possible correlation between cone production and mountain aspect. The cones production was also not correlated with aspect.

Reviewer #2: This paper describes an important evolutionary adaptation: behavioural variability in the amount of food cached by the endangered Mount Graham red squirrel (Tamiasciurus fremonti grahamensis). Authors tested the variation in cache size in response to microclimate variation, based on the following parameters: relief, vegetation (tree stand density), food availability and squirrel sex. Based on assumptions and predictions obtained from previous studies scientists expected that squirrels should locate bigger caches in areas:

1. Where the conditions to cone preservation are better and spoilage is expected to be reduced (colder areas).

2. With colder microclimates: in higher elevations and in slopes facing the northeast.

3. With higher cone availability.

To verify these hypothesis authors selected 40 middens occupied by squirrels distributed along different altitudes, relief, and mountain aspect. They visited each selected midden six times between the fall and winter of 2015.

Results of the study show, that much more pits were excavated in the northeast facing slopes. Surprisingly, the density of trees negatively affects the cache volume on southwest slopes only. The sex of the squirrel also affects the number of pits in the squirrel midden, males excavating more pits than females. Additionally, both sexes presented different responses to the mountain slope aspect (females excavating more pits on northeastern slopes, whereas male cache size did not vary with the slope aspect). Contrary to authors’ predictions the squirrel’s caching behaviour did not vary in response to midden microclimate variation.

However, although both the idea and conducted study are very interesting and worth publication, there are some aspects which require further explanation.

First of all, in my opinion (and as the authors mentioned in the discussion), the sample size may be not big enough for such multifactorial analysis. The second question is a type of distribution of studied middens – does it cover all habitat variability?

R: Indeed, due to the unexpected high mortality of squirrels during the winter, we worked always with the limitations of a small sample size in mind, choosing the less demanding statistical approaches, such reducing the number of variables before compare the models and never creating models with more than independent 2 variables. Despite our efforts, there is a limit to what is possible to do, and this is the reason why, in good faith, we added the warning in the discursion section (as mentioned by the reviewer).

About the distribution of the study middens, since the squirrel distribution is very limited, the sampling did cover most the habitat variability. We added a phrase in the manuscript about the subject. The middens coordinates are also provided in the supplementary material.

For me, the relationship between altitude, microclimate and the number of pits in the squirrel’s midden is the most important and missing information which may help us understand behavioural adaptation to the climate change. Authors should expect differences in caching squirrels’ behaviour in regard to differences in elevation gradient. In my opinion this aspect should be deeply analysed and discussed. Maybe squirrels territories were located at the same elevation – it should be disclosed.

R: The altitude of the middens varied between 2770 and 3270 meters (as mentioned in the “study area” subsection), which is almost the entire range of altitude of the population. However, Altitude was not “deeply analyzed”, because it was not a good predictor of squirrel cache size and was dropped in the variable reduction part of the analysis.

We added an entire paragraph to better explain the issue in the manuscript.

I am also wondering why authors analyzed differences between squirrels’ sexes? Did they expect any sex-related differences in catching behaviour and if so, there should be an explanation.

It would be good to introduce potential readers for studied species biology and describe briefly the subject of the study, including sexual differences, body mass and social system.

R: We added more details about how the squirrel sexual differences and why it is expected to affect the animal dependence on cached food.

When you try to consider what causes could lead to the obtained results – the lack of behavioural reaction on micro-climate variability – the good way is to analyze the used methodology. It is possible that daily air temperature, which was measured in the meteorological station differs slightly from the daily air temperature in other places, like middens. The same could be applied to soil temperature monitored by data-logger between one (the same) place and other places (e.g. middens).

Other explanation which the author discussed in the manuscript is that the most important for catching behaviour was the maximum, not the average air temperature.

R: Indeed, we agree with the reviewer, and it was the used approach. But we did not discuss about possible differences between the measurements between the meteorological station and the middens, since this would result in constant error along the middens, and therefore, it would be automatically corrected by the used methods. As the reviewer mentioned, we preferred to discuss about possible extreme events (such as temperature peaks), which would require a very different sampling method, and therefore, are beyond the original design of the present study.

Abstract

P2 L22 - …” in response to climate”… - should be “in response microclimate”

R: the phrase was changed to “in response to microclimate”

Materials and Methods

P6 L131 – 134

“We visited each midden 6 times between the fall and winter of 2015 to measure

variables of weather, relief, plant structure and estimate the volume of cones stored by

the resident squirrels (response variable). During the visits, we also record the sex of the

resident squirrels.”

Could you please explain how did you record the sex of the resident squirrels?

It’s unclear – Did you measure all of these variables during each of the six visits? If not please add the information when did you measure each variable. As I understood the temperature in all middens was measured on the same day.

How did you choose three pits from one midden for temperature monitoring?

How accurate was your equipment for the weather variables (handheld weather meter Kestrel 3000) and a digital soil thermometer (HANNA HI45-30), it’s possible that low accuracy can mask differences in temperature and humidity.

R: The sex of the resident squirrels was recorded by direct observation (using a binocular). It is a very easy and reliable method for sexing squirrels since they are usually up in the branches. We added the information in the manuscript.

We also add the information about when each variable was collected.

We didn’t think the equipment precision was a problem, since it is a very reliable equipment. The technical specifications can be accessed in the respective equipment websites (±0,27°C for the HANNA HI45-30 and ± 0.04C for the Kestrel 3000).

Discussion

In this part of the manuscript, there is no explanation of why sex should generate differences in squirrel’s catching behaviour.

R: The information was added in the methods section, as the reason of why we recorded squirrel sex.

Additionally, the authors’ hypothesis about different levels in predation risk between places with high and low tree steam density as an explanation in differences in squirrels’ catching behaviour seems to be unconvincing. First of all, it is hard to believe that predation risk can vary between different mountain slope aspects.

R: We thank the reviewer to point that the argument was not clear. We make some changes to make it clearer. Indeed, the argument is not intended to explain the different responses of males and females to the mountain aspect, but only in relation to tree density.

---

## [Editor Report · Decision Letter 1]

1 Oct 2019

PONE-D-19-16349R1

Does caching strategy vary with microclimate in endangered Mt. Graham red squirrels?

PLOS ONE

Dear Dr. Pereira Mendes,

Thank you for submitting your manuscript to PLOS ONE. After careful consideration, we feel that it has merit but does not fully meet PLOS ONE’s publication criteria as it currently stands. Therefore, we invite you to submit a revised version of the manuscript that addresses the points raised during the review process.

We would appreciate receiving your revised manuscript by Nov 15 2019 11:59PM. To enhance the reproducibility of your results, we recommend that if applicable you deposit your laboratory protocols in protocols.io, where a protocol can be assigned its own identifier (DOI) such that it can be cited independently in the future. For instructions see: http://journals.plos.org/plosone/s/submission-guidelines#loc-laboratory-protocols

We look forward to receiving your revised manuscript.

Kind regards,

Jesus E. Maldonado, Ph.D

Academic Editor

PLOS ONE

Additional Editor Comments (if provided):

I have gone over the response to the reviewer's comments and agree that the manuscript is much improved and that the authors have successfully addressed their concerns. However, I still noticed several editorial and grammatical errors that I made directly onto the revised version of the text in the attached pdf file. I suggest that the authors read and edit the manuscript carefully for grammatical and stylistic errors. This should be easy and quick to address.

---

## [Author Response · Author response to Decision Letter 1]

14 Oct 2019

The changes orthographic changes were addressed.

The figures were verified in PACE and changed when required.

---

## [Editor Report · Decision Letter 2]

25 Oct 2019

Does caching strategy vary with microclimate in endangered Mt. Graham red squirrels?

PONE-D-19-16349R2

Dear Dr. Pereira Mendes,

We are pleased to inform you that your manuscript has been judged scientifically suitable for publication and will be formally accepted for publication once it complies with all outstanding technical requirements.

With kind regards,

Jesus E. Maldonado, Ph.D

Academic Editor

PLOS ONE
---

## [Editor Report · Acceptance letter]

30 Oct 2019

PONE-D-19-16349R2 

Does caching strategy vary with microclimate in endangered Mt. Graham red squirrels? 

Dear Dr. Pereira Mendes:

I am pleased to inform you that your manuscript has been deemed suitable for publication in PLOS ONE. Congratulations! Your manuscript is now with our production department. 

With kind regards,

on behalf of

Dr. Jesus E. Maldonado 

Academic Editor

PLOS ONE